# OpenReview forum: "Explainable Rewards in RLHF Using LLM-as-a-Judge"
_ICLR.cc/2025/Conference — ICLR 2025 Conference Withdrawn Submission_

### Official Review · Reviewer_S2EC · 2024-11-03

**Soundness:** 2
**Presentation:** 2
**Contribution:** 2
**Rating:** 3
**Confidence:** 4

**Summary:**

This paper proposes an explainable reward model approach for RLHF using LLMs as judges. The authors generate multi-dimensional reward signals across specific task-related dimensions, such as helpfulness, truthfulness, and harmlessness, instead of traditional black-box reward models. The experimental results show that this explainable model improves alignment performance to black-box models while offering enhanced transparency and flexibility​.

**Strengths:**

The problem studied in this paper is significant for alignment. Also the topic of explainable rewards is interesting to the community.
The structure of this paper is well organized.

**Weaknesses:**

The paper lacks a detailed explanation of the reward model's multi-dimensional scoring system used during task evaluation. The authors mention using multiple dimensions to enhance interpretability but do not provide clear definitions for each dimension, specify their sources, or explain how many dimensions were ultimately included.

In addition, there is no direct comparison with SOTA models, and the dataset choices are not well-justified. In Section 2, the authors reference multi-objective approaches but do not provide a comparative analysis. And given that many methods use multi-objective and multi-agent techniques to enrich reward signals, the authors should discuss the rationale behind selecting specific dimensions, the reward generation mechanism, and how these differ in effectiveness from other approaches.

The limitations of the explainable reward dimensions and the impact of different dimensions on alignment performance are underexplored. While the method emphasizes transparency in reward modeling, there is little comparison to other interpretability-focused approaches, and the performance gap with existing methods is not addressed. Using GPT-4 only for evaluation and without a clear standard cannot serve as a trustful ``judge''. The qualitative analysis is also limited, missing deeper insights into unique semantic changes across different reward dimensions. Improving clarity, expanding comparisons, and providing a more robust analysis would strengthen the paper.

**Questions:**

Please include more comparative methods, such as traditional black-box reward models and other explainable reward modeling approaches. Also the authors may provide explanations and validations for the reasoning behind each dimension to justify their relevance and effectiveness in evaluating alignment performance.

---

### Official Review · Reviewer_Rdep · 2024-11-04

**Soundness:** 2
**Presentation:** 3
**Contribution:** 2
**Rating:** 3
**Confidence:** 4

**Summary:**

This paper proposes an explainable reward modeling approach for Reinforcement Learning from Human Feedback (RLHF) to enhance the transparency of language model alignment. The authors introduce a framework with LLM as a scoring model that generates interpretable rewards by scoring model responses along task-specific dimensions like helpfulness, truthfulness, and harmlessness. The experimental results show that the proposed explainable reward model achieves comparable alignment performance to traditional black-box models but with added explainability in specific RLHF tasks​.

**Strengths:**

1. The topic of explainable reward modeling for RLHF is relevant, and the focus on transparency addresses a crucial need in aligning AI models with human values.

2. The adaptation of task-specific scoring dimensions like helpfulness, truthfulness, and harmlessness makes the reward model more interpretable and tailored to evaluating language model outputs in sensitive applications.

**Weaknesses:**

1. For experiments, the paper presents an explainable approach to reward modeling, but it does not compare its method with recent state-of-the-art RLHF reward models, such as those using black-box or policy-based approaches. Please refer to:
https://arxiv.org/abs/2405.20053
https://arxiv.org/abs/2408.15240

2. In Section 3.3 BLACK-BOX REWARD SIGNAL GENERATION, it introduces dimensions such as helpfulness, truthfulness, and harmlessness, but it does not clearly define how these dimensions are quantified or differentiated during the evaluation.

3. The scoring mechanism for the dimensions' evaluation is vaguely described, and the thresholds for scoring levels are not provided. This limits the reproducibility of the experiments and the "explainable reward model" cannot explain how scores are assigned to responses.

**Questions:**

Can authors provide a deeper error analysis, particularly in scenarios where the model's rewards differ significantly from human evaluations? Meanwhile the paper could benefit from providing examples of cases where the model's rewards diverge from human evaluations, and an analysis of potential reasons for these discrepancies.
Also, for different alignment or RLHF datasets, should these dimensions maintain the same weighting? Additionally, the multi-dimensional reward approach appears to align closely with Multi-objective reward modeling.
For example: https://arxiv.org/abs/2406.12845
Could the authors provide evidence that their multi-dimensional reward model offers a clear advantage over traditional multi-objective reward approaches? I suggest that the authors discuss how their approach differs from or improves upon traditional multi-objective reward modeling, and address whether and how dimension weightings might need to be adjusted for different datasets or tasks.

---

### Official Review · Reviewer_gHBu · 2024-11-04

**Soundness:** 2
**Presentation:** 2
**Contribution:** 2
**Rating:** 3
**Confidence:** 4

**Summary:**

The authors of this paper aim to address the issue of dependency on human-annotated preference data in Reinforcement Learning from Human Feedback (RLHF) process. Specifically, they propose a new framework that uses an external large language model as a reward signal generator. This LLM independently evaluates scores for each relevant dimension and then employs a simple interpretable model (such as average scoring) to compute the final reward signal for use in the RLHF process.

**Strengths:**

1. The paper proposes a method that applies LLM to generate reward signals directly based on its evaluations of specific dimensions. This process avoids the potential data collection and training processes, thus reducing the costs of deploying RLHF in practical settings.

2.This method generates scores along specific, interpretable dimensions and uses an explainable model to combine these scores into a final reward signal. This enhances the explainability of the reward signals compared to black-box models that directly output the final reward.

**Weaknesses:**

1. This method heavily relies on the capabilities of the underlying LLM serving as a judge. If the LLM's understanding or output quality is suboptimal, it will directly impact the quality of the reward signals.

2. This method customizes evaluation metrics based on the characteristics of the task, using an LLM to generate scores for these specified dimensions. However, the paper needs to provide a systematic approach to developing these metrics. Considering that the scores generated by this method highly depend on these dimensions, this could lead to difficulties in transferring the technique to other tasks.

3. In the related work section, the authors specifically mention existing works aimed at reducing the need for human-labeled data in RLHF. However, the experiments in the paper only compare the results with a black-box reward model and do not include these methods as baselines for comparison. This weakens the comprehensiveness of the method's experimental validation.

**Questions:**

1. In the paper, LLMs are directly used as judges to generate reward signals. Suppose the LLM itself has inherent biases or inaccuracies in understanding. How might these issues be inherited through the reward signals they generate, and could this reinforce undesirable behaviors in the fine-tuned model? Exploring strategies to neutralize or compensate for these biases could be crucial for improving the fairness and accuracy of the RLHF process.

2. What are the best practices for designing prompts that effectively elicit the desired dimensions from an LLM? Systematic methodologies or frameworks are needed to guide the prompt engineering process and ensure the method's stable and effective performance across various tasks.

3. The method proposed in the paper integrates scores from various dimensions using a simple interpretable model. Could different ways of incorporating these scores impact the final RLHF outcomes? Research could discuss how different integration methods affect the method's performance.

---

### Official Review · Reviewer_BmKu · 2024-11-05

**Soundness:** 1
**Presentation:** 1
**Contribution:** 1
**Rating:** 3
**Confidence:** 3

**Summary:**

This paper proposes a novel framework for aligning Large Language Models (LLMs) with human preferences by introducing an explainable reward method in Reinforcement Learning from Human Feedback (RLHF). Instead of relying on black-box reward models trained on human preference data, the authors use an out-of-the-box LLM to generate reward signals based on predefined human value dimensions specific to a task (e.g., helpfulness, truthfulness, harmlessness). The LLM assesses responses along these dimensions using carefully designed prompts with clear rubrics, and the scores are combined using an interpretable model (such as a simple average) to produce an overall reward signal. The authors conduct experiments on question-answering and summarization tasks, demonstrating that their approach achieves performance comparable to traditional black-box RLHF methods while providing superior explainability and flexibility.

**Strengths:**

1. The proposed approach decomposes the reward signal into interpretable components based on human value dimensions. This approach may enhance transparency by making it clearer why a particular response receives a certain score, addressing the opacity of traditional black-box reward models.
2. While the experiments are conducted on two tasks, the underlying framework is designed to be generalizable across different tasks and domains.
3. The paper addresses important concerns in AI alignment by proposing a method that enhances transparency and reduces reliance on costly human data, contributing valuable insights to the ongoing discourse on ethical and effective AI development.

**Weaknesses:**

Although I agree with the necessity of a more transparent approach for RLHF, the paper does not provide a complete study. Specifically, some limitations in the justification of the proposed approach are as follows.
1. The paper proposes using Large Language Models (LLMs) to generate reward signals by scoring along predefined human value dimensions without providing a theoretical or empirical foundation for this approach. Although this paper refers to Zheng et al. (2023), it offers insufficient discussion on whether LLMs can reliably and validly act as judges in evaluating responses according to human values, particularly in the context of RLHF. A motivational experiment could be used to strengthen the justification for the proposed approach.
2. LLMs may involve biases, which can influence their evaluations and the generated reward signals. The paper does not address how these inherent biases might affect the reliability of the reward signals or how they are mitigated. The authors may consider incorporating debiasing techniques appropriate to the specific problem they are trying to solve.
3. A relatively minor but still important point is that the method uses a simple average of scores across different human value dimensions, implicitly assuming all dimensions are of equal importance. There is no theoretical or empirical justification for this assumption, and it may not reflect actual human preferences where some dimensions could be more critical than others. As far as I know, in social science, accurate identification and prioritization of reliable human value dimensions is a prerequisite to its use.

There are methodological or technical limitations as well.

4. Defining human value dimensions and creating detailed rubrics is resource-intensive. While the method does not demonstrate superior performance, the added complexity was not justified.

5. The approach relies solely on LLMs for both reward generation and evaluation, without incorporating human evaluators to verify the validity of the LLM-generated scores. This raises concerns about the method's reliability and whether it truly aligns with human preferences. While the paper evaluates the performance of the models primarily using GPT-4 as the evaluator without involving human evaluators to validate the results, this reliance on LLMs for both training and evaluation could introduce circularity and does not provide independent verification of the model's alignment with human preferences.

6. The experiments are conducted on only two tasks (question-answering and summarization), which limits the generalizability of the proposed approach. The applicability of the method to other tasks or domains remains untested and uncertain.

7. The paper does not analyze the consistency and reliability of the LLM-generated scores used for reward signals. Without assessing the variance and potential randomness in the LLM's evaluations, it's unclear how stable and dependable the reward signals are. I recommend the authors perform specific analyses or use metrics to assess consistency and reliability.

8. The paper does not report statistical significance tests to support the claims that the proposed method performs comparably to or better than the baseline. Without such tests, it's unclear whether observed differences are statistically meaningful or could be due to chance. The authors may conduct appropriate statistical tests or report relevant statistics to provide a more complete picture of the results and help assess the practical significance of any improvements.

**Questions:**

In relation to the weaknesses, please consider the following questions.
1. How can the authors theoretically or empirically justify the assumption that LLMs are capable of accurately and consistently evaluating responses along human value dimensions without human oversight?
2. How have the authors ensured that the LLM-generated evaluations align with actual human judgments, and why was human verification not incorporated?
3. What measures have been taken to identify and mitigate biases inherent in LLMs when they are used as evaluators, and how do these biases impact the trustworthiness of the reward signals?
4. Why were human evaluators not used to validate the performance, and how can the authors be confident that the improvements are meaningful from a human perspective?
5. Do the benefits of explainability and flexibility provided by the method outweigh the additional effort and complexity required to define dimensions and rubrics, especially when performance improvements are not significant?
6. Can the authors provide evidence or experiments demonstrating the method's applicability to a broader range of tasks and domains?
How have the authors ensured that the LLM-generated scores are consistent and reliable, and what analyses have been performed to assess their stability?

---

### Note · Authors · 2024-11-19

I have read and agree with the venue's withdrawal policy on behalf of myself and my co-authors.